# Distribution and Chemical Analysis of Pharmaceuticals and Personal Care Products (PPCPs) in the Environmental Systems: A Review

**DOI:** 10.3390/ijerph16173026

**Published:** 2019-08-21

**Authors:** C.R. Ohoro, A.O. Adeniji, A.I. Okoh, O.O. Okoh

**Affiliations:** 1SAMRC Microbial Water Quality Monitoring Centre, University of Fort Hare, Alice 5700, South Africa; 2Department of Chemistry, University of Fort Hare, Alice 5700, South Africa; 3Applied and Environmental Microbiology Research Group, Department of Biochemistry and Microbiology, University of Fort Hare, Alice 5700, South Africa

**Keywords:** pharmaceuticals and personal care products, endocrine disruptors, ultrasonic-assisted extraction, liquid chromatography mass spectrometry, solid-phase extraction

## Abstract

PPCPs are found almost everywhere in the environment especially at an alarming rate and at very low concentration in the aquatic systems. Many methods—including pressurized hot water extraction (PHWE), pressurized liquid extraction (PLE), ultrasound-assisted extraction (UAE), and micro-assisted extraction (MAE)—have been employed for their extraction from both surface waters and biota. Solid-phase extraction (SPE) proved to be the best extraction method for these polar, non-volatile, and thermally unstable compounds in water. However, ultrasonic extraction works better for their isolation from sediment because it is cheap and consumes less solvent, even though SPE is preferred as a clean-up method for sediment samples. PPCPs are in groups of—acidic (e.g., diclofenac, ibuprofen, naproxen), neutral (e.g., caffeine, carbamazepine, fluoxetine), and basic pharmaceuticals, as well as antibiotics and estrogens amongst others. PPCPs which are present in trace levels (ng/L) are more often determined by liquid chromatography-mass spectrometry (LC-MS), gas chromatography-mass spectrometry (GC-MS), and high-performance liquid chromatography-ultraviolent (HPLC-UV). Of these, LC-MS and LC-MS-MS are mostly employed for the analysis of this class of compounds, though not without a draw-back of matrix effect. GC-MS and GC-MS-MS are considered as alternative cost-effective methods that can also give better results after derivatization.

## 1. Introduction

Pharmaceuticals and personal care products (PPCPs) are over the counter therapeutic and veterinary drugs, ranging from analgesics and antibiotics to contraceptives and lipid regulators in addition to active ingredients in soaps, detergents, musk, bleaches, dyes, deodorants, shampoos, perfumes and hair cream, and skin and dental care products. Continuous introduction of these products to the environment, given their multiple disperse mechanisms and pharmacological activities may result in detrimental impacts on wildlife and humans [1,2,3,4,5,6]. Similarly, Ebele et al. [7] defined pharmaceuticals as inappropriate use of drugs and veterinary therapeutic medicines which are employed to check or cure animal and human diseases, whereas personal care products (PCPs) are utilized mostly to enhance the standard of every-day life. PPCPs are grouped as pharmaceuticals, and personal care products (PCPs). PPCPs comprise of chemicals that are biologically energetic, hormone-disruptive, and toxic [8]. Pharmaceuticals include antihyperlipidermis, stimulants (caffeine), analgelsics (ibuprofen, diclofenac, paracetamol, ketoprofen, naproxen), psychiatric drugs (carbamazepine, primidone), antibiotics (sulfamethoxazole, chloramphenicol, trimothoprin, ciprofloxacin), lipid regulators (gemfibrozil, bezafibrate, propranolol, atenolol, metoprolol), and antipyretic; while PCPs include synthetic musk fragrance (nitropolycyclic musks), antimicrobial compounds (triclosan), UV blockers (methylbenzylidene camphor), antioxidants and preservatives (phenols and p-hydroxybenzoic acid (parabens)) and insect repellents (*N*,*N*-diethyl-m-toluamide (DEET) [9], hormones (estrone E1, estradiol E2, ethynlestradiol EE2) [10]. PPCPs frequently detected in the environment include antibiotics, analgesics, steroids, antidepressants, antipyretics, stimulants, antimicrobials, disinfectants, fragrances, and cosmetics. This is based on their daily usage and consumption [11]. Yang et al. [12] in their research in advanced wastewater reclamation plant in Gwinelett County detected sulfamethoxazole, primidone, caffeine, and DEET frequently in the final effluent at concentrations of 10–100 ng/L. Moreover, Vlachagrammi and Athanasios [13] stated that steroids and non-prescription drugs are the most frequently detected. These include *N*,*N*-diethyltoluamide, caffeine, triclosan, copostanol, cholesterol, tri(2-chloroethyl)phosphate, and 4-nonylphenol.

In the last decade, environmental pollution of aquatic systems by PPCPs has been of increasing anxiety [14,15]. The rise in fate, effect and probability of exposure to danger by PPCPs has suggested the demand for dependable analytical method for quantification in diverse matrices. PPCPs have predominantly been detected at very low concentrations, although some are comparatively incessant owing to their constant usage and delivery into aquatic environment. As a result, aquatic organisms may possibly be subject to risk of these contaminants at a sub-lethal concentration over the time of their life phase [15]. According to Archer et al. [16], the existence and fortune of PPCPs in the surroundings are of growing prevalent significance as a result of their ubiquitous identity and reported consequence on wildlife, ecosystem, and potentially, humans. Increase in agricultural productivity has steered to increase the use of pesticides, of which most are persistent [17]. PPCPs have been marked the most recurring identified organic contaminants in the aquatic surroundings. Notable concentrations have been discovered in environmental water at a range from ng/L to hundreds of µg/L [18,19,20,21,22,23,24] and ng/L in municipal wastewater [25] and yet it is unclear whether this is posing danger to humans and wildlife anthropologically [16]. Besides, there is no regulation in place for this class of organic contaminants at the moment [26].

PPCPs are bioactive, they do not solubilize with ease or evaporate at ordinary temperatures or pressures, and so can enter with ease into the soil and water through sewage, bio-solids, and irrigation. They are harmful to both humans and animals. There is also indication that PPCPs are related to a number of ecological harm such as retarded maturity in fish and also in their metamorphosis [27]. Though the concentrations of single pharmaceuticals shown in evaluated water bodies globally are small and may be incapable of causing harm to the health of humans, constant vulnerability to a combination of such substances may cause disorder to the stability in the human body and intensify a high risk opposition to antibiotics, thereby constituting menace to the well-being of existing organisms; a burden that numerous scientists are presently exploring in other to find solution to it [5,28,29,30,31]. Some likely effects of PPCPs on living organisms have been reported, they include: increased feminization and masculinization of fish populations [5,32], nervous system damage, cancer, disruption of immune system, and the reproductive disorders [33]. They have been enlisted as “emerging contaminants” [11,26] and also as endocrine disruptors (EDCs) because they are capable of mimicking or altering various vertebrate endocrine systems pathways [34,35,36].

Garrison et al. [37] reported that the first discovery and publication that addressed pharmaceuticals in wastewater and effluents was given by US EPA, following this, Kummerer [38] stated that there is wide gap of data documentation on existence, out-turn, or action of PPCPs and their metabolites in waters. Management of pharmaceuticals and the treatment associated with them is not as inflexible in African countries compared to developed countries, and the present wastewater treatment systems are not intended for managing pharmaceuticals as pollutants [32]. Effluents of urban wastewater and receiving waters had been shown to contain numerous pharmaceuticals at low concentrations in developed countries where carbamazepine recorded the utmost concentration of 2.33 ng/mL in England [39,40,41,42,43,44,45] 0.21–2.62 ng/mL in Spain [30] 2.01 to 16.1 ng/mL in Saudi Arabia [46] and <1 ng/mL in Ireland [41]. Pharmaceuticals have been considered as emerging contaminants within the environment mostly as a result of their increasing usage, inappropriate discarding of unutilized or drugs that have expired and ineffectiveness of wastewater treatment plants (WWTP) to eliminate these contaminants completely [8,47,48]. Consequently, they find their ways into ground, marine, and drinking waters [4,44,49]. Their occurrences in the environment have been associated with harmful effects on the healthiness of humans and animals. Such effects include nervous system damage, cancer, disruption of immune system, and the reproductive disorders [33]. They have been enlisted as “emerging contaminants” [11,26] and also as endocrine disruptors (EDCs) because they are capable of mimicking or altering various vertebrate endocrine system pathways [34,35,36].

EDC is defined by the USEPA as an exotic medium that inhibits the synthesis, release, binding, transport, action, or annihilation of natural hormones in the body with the purpose of accountability for the conservation of reproduction, development, homeostasis, and/or behaviour [50]. Synthetic compounds largely regularly incriminated as EDCs involve pesticides, PPCPs, and industrial by-products [16]. PPCPs have various effects on aquatic organisms [51,52], deadly toxicity inclusive at incredibly elevated concentrations and changes in bacterial communities in aquatic ecosystems [53,54,55] with special significance on transience. Although, a number of outstanding sub-lethal effects—such as histological changes, behavioural effects, biochemical responses, and gene regulation—can arise at low concentrations [56]. Studies have shown the feminization of masculine fish and ubiquitous existence of anti-androgenic ligands in UK Rivers which collect effluents from linked WWTWs [57,58,59] also reported the distortion of male reproductive system in immature male alligators in diverse lakes located in Florida, USA owing to these pollutants [59]. Together with the distress about the prevalent disorder of human reproductive systems tending to several environmental unfriendly consequences such as breast cancer, ovarian cancer, and reduced sperm quality, global concerns were expressed concerning the probable clever disruption of the endocrine systems of wildlife and humans in the organizational window through development [33,60].

In Europe, there has been ban on chemicals commonly used in the production of PCPs which include phthalates and parabens. Rise in prescription for aging people with one chronic disease or the other; or youths having certain health challenges including neurobehavioural developmental disorders, has led to increased absorption of drugs in the USA [8]. Richardson et al. [61] reported that China was the world largest producer of pharmaceuticals in 2003 with annual production of 28,000 of penicillin (60% of world total production), 10,000 tonnes of terramycin (65% of world total production); and also ranked first among nations for the production of doxycycline hydrochloride and cephalosporins. Similarly, China is known for their legendary in the use of antibiotics, consuming about 70% of antibiotics used nationwide. Furthermore, Mutiyar and Mittal [62] reported that India is among the top five countries producing PPCPs. They are found increasingly in their surface waters notwithstanding their scarcity of water. Owing to this, humans and aquatic biota are exposed continually to these contaminants. Disappointingly, no needful action has been taken in this region to arrest the situation [62].

Examples of different kinds of PPCPs include:

### 1.1. Carbamazepine

Carbamazepine is among the class of antiepileptic/psychiatric drugs used to control seizure, mental illness and depression [63]. It is an anthropogenic marker [64] and has been considered as wastewater indicator in aquatic environment owing to its persistence [65,66]. According to Faigile and Feldman [67], 72% of CBZ oral dose are excreted in urine and 28% in faeces. It has been the most frequently detected pharmaceutical residues in water bodies [68], terrestrial environment [62], and WWTPs, in which little or no removal from wastewater has been widely reported, thus making it very important [68].

### 1.2. Erythromycin

Detection of this antibiotics in the environment has been of major concern because of its ability to alter the microbial structure and function, and also influence the development of antimicrobial resistance [69,70]. Its mechanism of action is through binding to the 50S subunit of the bacteria ribosome to impede protein synthesis [71,72]. It inhibits a definite cardiac potassium channel which appears to play a major role in cardiac rhythm regulation in the early embryo [73]. Erythromycin A, a 14 membered ring is the first macrolide, a class of antibiotics to be used to treat human infections [74]. Erythromycin has been reported in Asia, North America, Europe, China, and the UK and limited data has been reported in Africa, Eastern Europe, Central and South America [75].

### 1.3. Parabens

They are alkyl esters of p-hydroxybenzoic acid and are used as antimicrobial preservatives in consumer products including PPCPs with methylparabens, ethylparabens, and propylparabens being the major ones found in food stuffs [76] as a result of their high solubility in water, stability, low production cost, and relative safety when used [77,78,79]. Studies have shown that cosmetic products are its main source of exposure to human [78,79]. Its environmental pollution is of concern having been detected in human breast tumour as reported by Darbre et al. [80], sperm DNA damage in men [81], currency bills and sanitary wipes [82], WWTP [83,84], PCP [85,86].

### 1.4. DEET

DEET was developed in 1946 in the USA to guard the troops deployed from mosquito bites and related diseases [87,88], and thereafter made available to the public in 1957 as active ingredient of insect repellents. This can be found in spray, lotion for skin, in clothes and nets to guard against diseases transferred by vector bites and in prevalent areas [87,89] and antifeedant [90]. Merel [91] reviewed that DEET is one of the most commonly detected organic chemical contaminants in water.

### 1.5. Triclosan

Triclosan has since 1968 being used in consumer products such as antiseptic, disinfectants, cosmetics, preservatives in clinical settings, and also incorporated in medical devices, plastic materials, kitchen utensils and textiles as reported by Bedoux [92]. It has been detected in fish [93], human samples (urine, breast milk, and serum), because of its high lipophilicity, and also in sewage effluent, river, surface water, and sediment [94]. It is highly hydrophobic, biodegradable, photo-unstable and reactive towards chlorine and ozone.

### 1.6. Caffeine

Caffeine is one of the world’s most consumed stimulants. Its source is mostly from beverage and food especially coffee [95]. The leaves can be seen as herbicides as its plant prevents the growth of other surrounding plants. It has been used as insect repellent because of its bitter taste, and as toxicant at high dose [96]. It can also increase pollination to enhance reproduction [97]. There has been a report on its cardiovascular, behavioural, and reproductive effects [98].

This paper therefore seeks to review the characteristics, distribution, mobility, and physicochemical properties of PPCPs in environmental matrices, human samples and biota; and also to assess the available techniques for their analysis.

## 2. Characteristics of PPCPs

There has not been proper report about environmental risk of exposure to PPCPs, however, they possess the characteristics of chemicals that give a major concern with adverse effects on aquatic lives. These characteristics include persistence, bioaccumulation, and toxicity. Some PPCPs such as steroids, non-steroidal pharmaceuticals and a number of PCPs are endocrine disrupting [7]. Many PPCPs are persistent and hence, have been potentially risky when discharged into the environment [16].

### 2.1. Persistence

They are not easily eliminated from the typical water treatment operations, thereby posing threat to aquatic lives exposed to them [99]. Occurrence, treatment, and toxicological relevance of EDCs and pharmaceuticals in water [100]. Houtman [101] and Daughton and Ternes [11] also added that some of the PPCPs have restricted lifetime in the environment and are not necessarily persistent but are repeatedly used and released to the environment even after the process of biodegradation, photodegradation, and sorption; and so classified them as “pseudo-persistent”. Several of these PPCPs have been reported to be tenacious during wastewater treatment thus, constituting hazard when discharged into the environmental waters [47,67,102]. Similarly, Vieno et al. [103] stated that biodegradation, sorption, photodegradation, and sedimentation can also remove PPCPs in the aquatic environment. Consequently, Lam et al. [104] reported that half-lives of PPCPs ranges from 1 day with acetaminophen, to 82 days with carbamazepine in sun-drenched microcosms of bare pond water and normal autoclaved water, and that tested pharmaceuticals did not breakdown in the gloomy control microcosms, thereby signifying that photo degradation, not biodegradation, may possibly be a restricting factor in the perseverance of PPCPs.

### 2.2. Bioaccumulation

There is still lack of comprehension of kismet and bioaccumulation of PPCPs in the environment and their outturn on ecosystem function [105,106]. Bioaccumulation of PPCPs is affected by pH, presence of surfactants, and also suspended solids [107,108,109]. Shraim et al. [46] made an inference that not many health threats can be related to long term vulnerability to a single drug but with disclosure to a large number of pharmaceuticals, their metabolites, and transformation products, even at small concentrations. Muir et al. [110] repeatedly detected carbamazepine, metoprolol, and sulfamethoxazole in water but not in fish plasma indicating swift biotransformation.

### 2.3. Toxicological Effects and Health Risks

Antipyrine is one of pharmaceutical active compounds easily detected in aquatic organisms, capable of causing organ deterioration because of its cytotoxicity towards mucosa and lungs on long term exposure [111]. Though pharmaceuticals are not as tenacious as other organic compounds, they have been found in concentration ranging from mg/L to ng/L depending on the compounds, part of the environment in which they are determined, and degree of application of the substances in the study area. Low concentrations of these contaminants have proven to have potentials to induce unfavourable effect on lives in the ecosystem. The established risks include the development of microbial opposition to antibiotics on exposure to hormones [32]. Similarly, Wang et al. [112] and Snyder [100] reported that PPCPs contain various groups of compounds capable of generating potential risk on aquatic lives by inducing fish’s nephridial tissue necrosis, affecting the growth of algae and duckweed, intensifying the microbial resistance to antibiotics. They further stated that the most poisonous endpoint is not curative endpoint, but relatively the carcinogenic side effect and also life-threatening toxicity at an exorbitant temperature and alteration in bacterial communities in an aquatic ecosystem [53,54,55]. Although, at low temperature, behavioural effects, biochemical reactions, and gene management can also arise [56].

Pharmaceuticals being biologically active may have effect on non-target organisms at low concentration in terrestrial and aquatic environments. Chronic toxicity and possible insidious environmental effects are barely known, though acute toxicity effect on non-target organisms have been studied. Acute toxicity to aquatic organisms is improbable to occur as the measured concentration in the terrestrial environment is 100–1000 times higher than the residues found in aquatic environment. It is only salient in case of spills [113] or specifically for organisms inherent in effluent-dominated system [107]. PPCPs are not likely to cause acute toxicology since they appear in low concentration. The ecotoxicity, bioavailability, and duration of exposure to non-target organisms is a guide to evaluate the strength of these compounds in the environment [114], and so may be ignored in natural aquatic habitat [115]. Furthermore, life-cycle analysis is scarcely reported except for 17α-ethynlestrdiol [116] which has shown some estrogenic effect in reproduction on fish at very low concentration [117]. There is need for concern about feminization and masculinization of hormone and xenoestrogens, synergistic toxicity from complex mixtures at low concentration, possible invention of opposition strains in natural populations and other probable interests for human health; though the environmental toxicology is not clear [67]. Dussault et al. [118] investigated on four PPCPs which are carbamazepine, atorvastatin, 17α-ethynlestrdiol and triclosan and deduced afterwards that triclosan was the most toxic followed by 17α-ethynlestrdiol, atorvastatin, and carbamazepine in that order. Similarly, Delorenzo and Flaming [119], also carried out an investigation which showed that of six PPCPs (triclosan, fluoxetine, simvastin, clofibric, carbamazepine, and diclofenac), only triclosan produced toxicity at usual environmental concentration.

Rise in the levels of PPCPs such as penicillin and zithromax in the receiving waterbodies may eventually result in the increase in resistance to antibiotics [69,120]. There is no available data from risk assessment and ecotoxicology of most occurring emerging contaminants, so it is difficult to envisage their health implication on humans, terrestrial and aquatic organisms, and ecosystem. Furthermore, PPCPs occur in a very low concentration which is incapable to cause acute effect. However, health effect as a result of long-term exposure cannot be avoided [13]. Similarly, Becker and Stefenakis [121] reviewed that it is difficult to evaluate the exposure effects of PPCPs in human health, following that PPCPs are detected at sub-therapeutic concentrations. Most times, they appear in a mixture with other compounds making it impossible to fix which among the contaminants is the problem especially when chemicals form a compound and also because the persistence of PCPs is comparatively short and may not be detected.

## 3. Distribution and Mobility

Owing to the low volatility of PPCPs, their distribution will occur via water bodies and food chain dispersion than through the atmospheric environment. The polar and non-volatile nature of a huge number of them will as well limit their disappearance from the aquatic media. Volume of PPCPs produced will also to a great level dictate the extent of their distribution in the environment [7,11,122]. These classes of chemicals have great tendency for long range transport in the environment, subject to their physical and chemical properties, as well as the characteristics of the receiving environment. PPCPs are usually by adsorption found in sludge samples, and subsequently in the environment when the sludge is used as fertilizer on farmlands [123]. The presence of these chemicals in the groundwater, possibly due to leaching from biosolids applied to farmlands, or treated wastewater employed for irrigation has been reported [124,125,126]. This can in turn be taken up by plants, which now becomes a major way of human exposure by dietary intake [127,128]. They can also enter surface water through runoff from contaminated farmlands, biosolids or from landfills, and as a result presents a risk to both aquatic and public lives [7,11,13].

Generally, the movement of PPCPs across different environmental media is dependent on the sorption behaviour of the compound of interest in plants, soil, and the water-sediment system [129]. Sediment is considered the primary sink for the organic pollutants by sorption [130] and they may eventually be released back to the aquatic environment after several accumulations in the sediment. PPCPs such as sulfamethoxazole, carbamazepine, triclosan, and ciprofloxacin are reportedly more persistent in the sediment than water [131,132]. Continuous release of these organic chemicals from sediments to overlying water is therefore possible, and this may adversely affect the benthic organisms which are continuously exposed to PPCPs found in the sediments, interstitial and overlying waters [7,133].

## 4. Physicochemical Properties of PPCPs

### 4.1. Temperature, Polarity, and Stability to Heat

They are moderately polar, non-volatile and not stable to heat. It is therefore difficult to use GC for their analysis without prior derivatization [15,134].

### 4.2. pH

Pharmaceuticals vary from acidic (carboxylic acids, e.g., diclofenac, bezafibrate) to basic (secondary amines e.g., paroxetine, bisoprolol). Overall methodologies discriminating in opposition to interferences should be established to investigate the major environmentally pertinent pharmaceuticals [135].

### 4.3. Solubility

Majority of the pharmaceuticals are extremely soluble in water and consequently retain solubility in the aqueous phase while the ones with low solubility remain insoluble with solid materials in wastewater ([102,136,137]. They dissolve readily in aqueous form and do not vanish at ordinary temperature or pressure, and so find their way into the water bodies through treated sludge (biosolids), sewage, and irrigation with domesticated waters [138,139].

### 4.4. Volatility

Volatilization is the interchange of contaminants between water and gas phases. Generally, PPCPs have low volatility [119]. They are either non-volatile or semi-volatile, only few of them are volatile [15]. Their runoff from the aquatic sphere being principally spread in the course of aqueous transport and food chain is being prevented by their decreased volatility [11].

### 4.5. Sources of PPCPs in the Environment

Most of the components of PPCPs are man-made and are gotten from human activities. They do not occur in nature except few of them such as caffeine which is of plant origin, produced by over 60 plants. PPCPs get to marine environment through wastewater and polluted surface water, septic tanks, livestock breeding, landfills, and sewer leakage [137,140]. Suppes et al. [141] reported that PPCPs can be found in swimming pools through fill water and human impact like urine, sweat, swim wear, and body surface. They enter the environment through humans, veterinary consumption, excretion, bathing, discarding of unutilized products, flushed down toilets, wastewater treatment plants, surface runoffs and leaching [16,142,143], residues and wastes from hospitals, wastes from pharmaceutical industries, use of illegal and veterinary drugs and agribusiness [29,37,40,144]. Although, they can also be detected at non-point sources especially in surface waters at small concentrations as ng to µg/L [11]. It is however uncertain if this low concentration can cause unsatisfactory physiological consequence in wildlife and humans [16].

## 5. Sample Collection, Storage, and Preparation

Sample preparation performs an important task in analytical methodology. Most analytical instruments are inadequate to manage matrices directly, and some kinds of pre-treatment are needed to extract and isolate the analytes [145]. Sample preparation is taken to be the major contaminating step of analysis, since it normally needs the utilization of organic solvents [146].

According to [147], appropriate collection and preservative procedures are required to make certain sample probity and retain the steadiness of the analytes pending investigation. Data from inappropriately collected and preserved samples bring about misconception of the existence and outcome of the compounds being investigated. Emerging contaminants such as PPCPs and steroids mostly exist in drinking and surface water at ng/L levels. Amber bottles were discovered to have minimal consequence on concentrations of target analytes, whereas elevated density polyethylene bottles can greatly interfere. Preserving with sodium azide at 4 °C proved to enhance the steadiness of the target analytes in their study. Hence, amber bottles were used with sodium azide and ascorbic acid which preserved all the analytes at 4 °C for 28 days, except two were considered the most appropriate. Englert [148] also reiterated the need for sample collection using amber glass container in accordance with standard practices.

For aqueous samples, water that flows effortlessly is collected as the grab sample. Acidic and basic samples are collected in 1 L bottles. The bottles should not be rinsed before sample collection. Where residual chlorine is found, 10 mg sodium thiosulfate per litre of water is added. Ascorbic acid can be used as preservative for some pharmaceuticals, however, it has not been assessed for all of them. If aqueous sample is not extracted within 48 h of collection, pH should be adjusted to 5.0 to 9.0 with NaOH or H_2_SO_4_ solution. For solid samples, grab sampling is more frequently employed. Adequate amount of 10 g solids is collected and maintained in the dark <6 °C from collection point to the laboratory and then stored in the dark less than −10 °C until analysis. Extraction from aqueous, bio-solids, mixed phase, and solid samples should commence within 7 days (48 h is strongly recommended) as some may degrade promptly. The extract should be analysed within 40 days of extraction. Freezing of samples is recommended to reduce degradation, and extraction should be carried out within 48 h of defrosting [148]. Standard methods however suggested that extracts should be analysed as soon as possible, not exceeding 28 days after preparation [149].

Ort et al. [150] stated that errors in sampling can lead to over interpretation of constant data and eventually, incorrect deductions. On the determination of sewer type, catchment size, sampling structure, substance of scrutiny, and reliability of analytical method, preventable sampling relics can vary from ‘insignificant’ to 100% or more for various compounds even in the same study. Sampling errors and biases can be minimized significantly by selecting fittingly sampling mode and frequency.

## 6. Environmental Matrices of PPCPs

### 6.1. Wastewater, Sewage, and Sludge

Sewage treatment plants (STPs) is the major source of PPCPs in the environment [11]. Antibiotics is the main abundant group of pharmaceuticals found in STP of China having concentration up to 1730–7910 ng/L for influent sample and ND–94960 ng/L for effluent samples as reported by Liu and Wong [10]. Some of the greatest analytical conflicts to a thorough analysis of sewage sludge require managing the big negative surface charges and interstitial scope that gives multiple active sites for charged compounds, and the clean-up step for removing the magnitude materials (e.g., surfactants, fats, proteins) that are co-extracted with the pharmaceuticals [151]. During wastewater treatment, alteration of PPCPs possibly will take place based on the physicochemical properties of the compounds and the state of the WWT. PPCPs may possibly be totally ruined, or partly modified to metabolites or in a few occasions left unaffected during the process [151]. It is essential to know that the degradation or elimination of the parent analytes during WWT does not certainly denote the elimination of toxicity, it is anticipated that a significant number of alteration products whose toxicity and tenacity are unknown might still be found in the last effluent together with the receiving water bodies [109].

### 6.2. Surface Water and Sediment

PPCPs, after they are ejected, washed off from human body, or directly released to the sewerage system, go into surface water majorly through improper treated wastewater effluent. Most of them are regularly found in river water with concentrations of single μg L^−1^ and their levels depends majorly on the level of water dilution as a result of rainfall [152]. Various groups of PPCPs were investigated for their existence in surface waters. The degree of their concentrations in the aqueous environment is dependent on a variety of parameters such as efficiency of treatment of wastewater, meteorological conditions (mainly rainfall), proximity to wastewater plants, and geographical position [106,153,154,155,156,157,158,159,160,161,162,163].

PPCPs may go through diverse environmental processes, majorly photolysis and adsorption by sediments. A good number of antimicrobial media, such as triclosan, seem to be adsorbed and trapped in the pores of sediments and subsequently decrease its solubility. The course of adsorption can be influenced by pH [164]. Elimination of PPCPs can be achieved by adsorption from water bodies, though, PPCPs in the sediments (being their sink) may possibly be discharged back into the aquatic environment [165]. Aqueous photolysis is a possible way of removing tetracycline, although the process could still be improved by high pH [166]. Nonetheless, these contaminants are usually found in surface water sediments at a level much lower than in sludge samples [167].

### 6.3. Soil and Drinking Water

PPCPs can be introduced into soil by applying livestock wastes as fertilizers, sludge land application or landfill, and domesticated water irrigation. Contamination in soil may possibly be built-up in plants or drift through soil wholly or altered and get to ground water, finally leading to contamination of the source of drinking water. Veterinary antibiotics subsisted in the soil from organic vegetables farmland fertilized with livestock wastes from Tianjin in concentrations up to 2683 ng/g (dw) [168]. Lin and Gan [164] discovered some drug variety, like diclofenac and ibuprofen that showed inadequate adsorption with persistence under anaerobic condition. The leaching of PPCPs may possibly be influenced by chemical characteristics (such as pKa values), salinity of the irrigation water and soil properties (e.g., organic matter and clay content) [169,170]. For instance, Qiao [171] detected four kinds of PPCPs approximately 1 ng/L in drinking water in South China.

### 6.4. Biota

Global studies have associated the disclosure to effluents of WWTP having PPCPs with harmful influence on the marine organisms’ reproduction [172]. A study also showed bioaccumulation of a combination of estrogenic pollutants in fish tissues, thus leading to the incorporation of vitellogenin and most likely advancing to feminization of wild fish inhabiting UK Rivers [172]. A national pilot research in the US, evaluated build-up of PPCPs in fish sampled from five effluent-controlled rivers which receives express release from wastewater treatment in Florida, Pennsylvania, Illinois, Arizona, and Texas. The research showed the existence of galaxolide and tonalide in fish fillets at each effluent-dominated site with maximal concentrations ranging from 300 to 2100 ng/L and 21–290 ng/L in that order. The pharmaceuticals discovered both in liver and fillets comprise; norfluoxetine, diphenhydramine, diltiazem, sertraline, fluoxetine, gemfibrozil, carbamazepine, with sertraline the most plentiful at maximal concentrations of 19 ng/L in fillet and 545 ng/L in liver tissue [173]. Also, the harmful decrease in the population of vulture in Pakistan has been partly accredited to dietary vulnerability of vultures to diclofenac-treated livestock. Particularly, the diclofenac was discovered in the kidneys of all 25 vultures that died of renal breakdown at the concentrations of 0.051–0.643 μg/g [174]. Li et al. [175] also discovered 13 antibiotics in majority of the investigated hydrophyte samples (aquatic plants) such as *Ceratophyllum demersum* (Cer), *Salvinia natans* (Sal), *Hydrocharis dubia* (Hyd); 4 crustacean species including river snail (*Viva parus*), crab (*Eriocheir sinensis*), lobster (*Palinuridae*), shrimps (*Macrobrachium nipponense*); and 7 fish species: yellow catfish (*Pelteobagrus fluvidraco*), loach (*Misgurnus anguillicaudatus*), and topmouth gudgeon (*Pseudorasbora parva*), among others from Baiyangdian Lake, China. The respective concentrations of antibiotics in Sal, Hyd, and Cer were 1769 μg/kg, 129 μg/kg, and 253 μg/kg [175]. Other researches have also reported the detection of ciprofloxacin concentrations in the aquatic plant (*Echinodorus amazonicus*) as high as 795 μg/kg [45,176].

Liu et al. [177] also showed concentrations of steroid estrogens at 11.3 ng/g dry weight (dw) in species of undomesticated fish such as silvery minnow, crucian carp, and carp from Dianchi Lake in Southern China. Liver showed the largest estrogen build-up, followed by gills and muscle. Also, Ali et al. [178] detected caffeine. Methylparaben and carbamazepine in concentration 41.3, 44.3, and 1.7 ng/g (on a dry weight basis) respectively. Presently, little is investigated regarding the PPCPs levels in biota generally. Not many studies have evaluated PPCP residues in mammals, fish, and birds. There is also a paucity of information on the possible trophic amplification of these analytes or the effect of prenatal exposure on the probable conveyance of PPCPs to eggs of birds and other nascent wildlife [7]. Some PPCPs have revealed to build-up in fish tissue sampled from surface waters accepting discharges of effluent [173,174,175,176,177,178,179]. PPCPs have been determined in diverse tissues of fish such as blood plasma, brain, fillet, and liver [18,111,173,174,175,176,177,178,179,180,181,182,183,184]. For example, the pharmaceuticals sertraline (STL), fluoxetine, carbamazepine (CBZ), diphenhydramine (DPH), gemfibrozil, and diltiazem, and PCPs triclosan, tonalide (AHTN), and galaxolide (HHCB), were currently shown in fish from five US rivers in the US EPA’s national pilot research of PPCPs in fish tissues [182]. Prior PPCPs studies have determined synthetic musk [185], alkylphenols and their monoethoxylates [186], and triclosan and one of its metabolites [185] in the German Environment Specimen Bank (GESB), from fish tissue samples collected from 1994 to 2003.

## 7. Extraction Techniques

Vogues in isolation of analytes include reduced solvent consumption, improved extraction throughout, increased recoveries, and better reproducibility [146,187]. According to Nieto et al. [188] and Pérez-Lemus et al. [189], several extraction techniques have been developed and used in investigating PPCPs in environmental matrices like sewage sludge. These methods are based on liquid partitioning [190], conflict extraction and chromatographic analysis; they include Soxhlet, ultrasonication (USE) [191,192,193], and supercritical fluid (SFE) extraction methods [194]. Recently, environmentally friendly extraction methods like microwave-assisted extraction (MAE) [2,195,196] and pressurized liquid extraction (PLE) [186,197,198] have become prominent and apparent as effective means of increasing automation, reducing extraction time and minimizing the amount of organic solvent [199]. Concerning modern extraction methods (e.g., PLE and MAE), their small extraction times (5–45 min), less solvent consumption, reduced cost of equipment and simplicity of operation are their major advantages over classical methods. The extraction and analytical techniques are summarised in Table 1.

### 7.1. Solid-Phase Extraction (SPE)

SPE is the major routinely utilised method [200,201,202,203,204] for extracting pharmaceuticals from water samples. Oasis HLB, with its hydrophilic–lipophilic balance, is mostly utilised for pharmaceuticals extraction with a broad range of polarities and pH values [205,206]. Oasis MCX, a mixed-mode resilient cation-exchanger, consequently gives both ion-exchange and reversed-phase retention and can adsorb neutral, polar, non-polar, and cationic compounds synchronously from aqueous media. MCX has been outstandingly used to extract an extensive range of pharmaceuticals and synthetic hormones from water matrices [152,207,208]. Many researchers extract PPCPs from aqueous sample using SPE except for few such as Yu et al. [209], who used the same SPE for the extraction of the analytes of interest from sludge as shown in Table 1. The use of SPE for aqueous sample is in accordance with EPA [148].

Similarly, work done by Shraim et al. [46] showed that Oasis MCX cartridges which was used for extraction of PPCPs can retain acid, base and neutral drugs in solution depending on the functional group(s) it possesses and on the pH of the solution, and has shown to give high overall recoveries. In the study, drugs were divided into two groups: “acidic and neutrals” and “basics”, this was to improve the sensitivity of the MS detector and to avoid any complication during analysis [41,210].

### 7.2. Pressure Liquid Extraction (PLE)

PLE utilizes pressure and temperature without extending to evaluative time [211] and has more superiority over traditional methods (such as Soxhlet and USE). Study conducted by Gobel et al. [197] also preferred PLE over USE in extraction of several antibiotics including sulphonamides (sulfapyridine, sulfamethoxazole), macrolides (azithromycin, roxithromycin, clarithromycin), and trimethroprim, in dried samples of activated and digested sewage sludge. These advantages include good recoveries [190], reduced extraction time, reduced solvent utilization and extra extract filtration, achieved by adding inactive material to the extraction [197,212]. PLE also has an advantage of not being restricted to extraction solvents that can absorb microwaves over newer technique like MAE. However, PLE has got some shortcomings of having selectivity not as high as might be needed towards analytes throughout extraction causing many interferents to be co-extracted, based on the kind of sample. Also, the analytes are occasionally diluted, mostly when an elevated number of revolutions is used [213]. To prevent this dilution by the interferents, SPE using Oasis HLB, Oasis MCX, and Strata-X cartridges, being the most common technique for clean-up should be utilized [190,197,214], with several sorbents. SPE also pre-concentrates aside cleaning of analytes. Gel-permeation chromatography (GPC) is another technique used for clean-up [215].

### 7.3. Ultrasound Assisted Extraction (UAE)

UAE is a cheap and effective substitute to conventional extraction methods (e.g., Soxhlet that needs extensive processing time of 4–24 h and high volume of solvent). It also extracts thermal unstable analytes that may possibly be transformed in the operational conditions of Soxhlet. In addition, cavitation expanses the polarity of extractants and analytes, hence improving extraction. The method can be utilised with whichever solvent. Extraction of analytes is mainly dependent on the solvent’s polarity, the nature and uniformity of the sample, the ultrasound frequency and the sonication time [33].

Employment of ultrasounds is a likely means for extraction effectiveness enhancement. Ultrasonic field cites numerous occurrences that may definitely influence the operation kinetics: decrease in viscosity, micro-floats and in the case of appropriately described parameters, also cavitation and rise of processed medium temperature. Hence, USE finds its wide utilization for isolation of several organic substances [216,217,218,219,220,221,222,223] though not reproducible [224]. The ultrasound activity is indivisibly linked with heat liberation in the investigated medium. The occurrences accountable for an addition of temperature in the managed material are: absorption of ultrasound energy friction of boundary and interface surfaces, and, in the case of high extremities, cavitation [225]. Nearly 75% of acoustic energy conveyed by ultrasonic converter may possibly be eventually modified into thermal energy of a system sorted out [226]. The relevance of UAE to the investigation of emerging contaminants in environmental solid matrices (e.g., sediment, soil, and sludge) has gained attention, as revealed in the several studies depending on UAE published articles in the recent time. Techniques for sludge samples are not as many as those for soil or sediment because of the complication of sludge. Parabens and UV filters have been effectively extracted from sediment and soil by means of UAE in small columns with acetonitrile or ethyl acetate, correspondingly, as extraction solvent, and SPE as a clean-up step [227,228]. A broad variety of analytes have been researched by means of ultrasound assisted extraction (USE). For example, applying it to pharmaceuticals and EDCs [191,229,230], antibiotics [231], and pharmaceuticals and personal care products (PPCPs) have been reported. However, equivalent solvents to those employed in Soxhlet are used during USE, both the extraction solvent volume and the time of extraction are decreased to 4–60 mL and 10–30 min, respectively [232]. It is an adopted method with acetonitrile for solid samples according to EPA [148].

### 7.4. Pressurized Hot Water Extraction (PHWE)

PHWE is similar to PLE but employs the use of water as the extraction solvent at elevated temperature and lower pressure [233,234]. The report in the literature contains only few examples of PHWE pertaining to emerging and persistent organic compounds in a diversity of matrices [235]. The major determinant that influences extraction effectiveness during PHWE is temperature; since high temperatures, low surface tensions, and low viscosities are achieved at high diffusions. Also, the compounds’ vapour pressure rises at elevated temperatures and hence, thermal desorption from the solid matrix takes place. Although, hydrolysis, degradation or oxidation of the target compounds can as well take place at elevated temperatures. pH of the water phase is another variable worked on during PHWE when working on analytes with acid–base properties, given that more of the charged part solubilizes in the water-phase and improved extraction effectiveness can be gained under those conditions. Other variables like those considered when discussing PLE—e.g., the number of cycles and the flush volume—are as well developed during PHWE [236]. PHWE has been successfully utilised for the analysis of non-steroidal anti-inflammatory drugs (NSAIDs), diclofenac, ketoprofen, naproxen, and ibuprofen [236].

### 7.5. Dispersive Liquid–Liquid Micro-Extraction (DLLME)

DLLME comprises of extraction with a water-immiscible solvent and a disperser that are promptly injected into the aqueous solution consisting the analytes under scrutiny, and leads to the generation of an unclear solution, separated subsequently by centrifugation. DLLME has gained unique awareness for the reason that its short extraction times, clarity, and high enrichment factors (EFs) for low water volumes. The application of sonication to this method has brought about increase in its significance in recent years because ultrasound-assisted dispersive liquid–liquid micro-extraction UA-DLLME gives a surged rate of mass transfer of analytes from the aqueous phase to the filmy extracting droplets.

The recovery of analytes in UA-DLLME is dependent on various factors, like the amount of sample, the extraction and disperser solvents and their volumes, the ionic strength or the pH. All extraction solvents employed in the investigation of emerging contaminants were chlorinated hydrocarbons (e.g., dichloromethane, chloroform, and carbon tetrachloride). Concerning the disperser, methanol, and acetone were the solvents mainly generally utilized. In the case of PCPs, UA-DLLME was useful for the concurrent determination of parabens and bisphenols [237].

### 7.6. Microwave Assisted Extraction (MAE)

MAE, also known as microwave-assisted solvent extraction (MASE) is one of the green solvent extraction methods which have been developed to eliminate the obvious pollution of releasing solvent into the environment which is applicable to traditional solvent extraction and technique for the valid investigation of emerging organic contaminants, validating its addition regarding regulatory environmental field [183]. It is a method of employing microwave energy to heat sample solution so as to separate analytes from the sample matrix into the solvent [145]. MAE is simple to apply and rapid, and gives good extraction effectiveness equivalent to or greater than those acquired with classical methods (i.e., Soxhlet or LLE) and other more current techniques (i.e., SFE or PLE) [238]. MAE can be static, a process whereby the sample is submerged in the extraction solvent and illuminated by microwave, an aliquot of the extract is aspirated by a pump for additional treatment subsequently; or dynamic (DMAE), a process whereby the extraction solvent constantly passes through the sample. DMAE can be either closed (performing an extraction in a recirculation system, the aliquot of the extract is aspirated at completion subsequently) or open (having the sample constantly exposed to fresh solvent, the extracted analytes are subsequently transferred to next analytical procedure).

MAE can also be focused microwave-assisted Soxhlet extraction (FMASE) (static-dynamic MAE) - which is dependent on the same principle as conventional Soxhlet extraction but employing microwave as auxiliary energy to accelerate the extraction process [145]. Among the major precedence of MAE, is the substantial decrease of extraction time and solvent consumption, and the option to carry out multiple extractions, increasing the sample all through. The suitable cost of the equipment should also be considered. The optimization of MAE situations is rather simple because of the small number of significant parameters (i.e., time, nature of the solvent, matrix moisture, power, and temperature in closed vessels). Other current applications of MAE reported novel concentration procedures depending on the utilization of ionic liquid (IL)-based surfactants and focused MAE for the pre-concentration and extraction from sediments of organic contaminants e.g., PAHs, alkylphenols, and parabens. Although microwave extraction is a routine method for solid samples, original micro-extraction approaches, indicating the utilization of surfactants and ILs, were currently conveyed for the pretreatment of water samples [238].

The use of MAE has stimulated growing curiosity, and has been generally employed in analytical chemistry because of its many superiorities. It has been implemented to resistant matrices such as drugs and pollutants in animal and human tissues, and to matrices collected at polluted sites where selection or characterization is needed for the reason of successive or on-going restorative work [239]. MAE, which can be performed with open or closed vessels, utilizes microwave energy to heat the sample–solvent mixture. The solvents for extraction accessible for MAE are normally restricted to those that take in microwaves (solvents with permanent dipole), though the utilization of solvent mixtures with and without dipoles proffers MAE applications to a variation of possible solvent mixtures [232].

From a green chemistry perspective, besides the cost and decreasing the energy input, the major advantages of MAE comprises of: significant reduction of solvent needed hence, reducing waste production; enhancement of extraction all through; and, decrease of sample amount needed [195,240]. Extraction time is much lower when implementing MAE, mostly as a result of the variation in heating implementation used by the microwave method and normal heating. With the later, a fixed period is required in heating the vessel before transferring heat to the solution; whereas the solution is heated directly with microwaves, so maintaining the temperature slope to the least and increasing the heating speed. Furthermore, MAE enables for obvious decrease inorganic solvent utilization and the probability of concurrently running multiple samples. MAE accordingly largely satisfies the minimal standard needed for modern sample preparation methods. However, it is a very desirable substitute to more conventional methods [146,201,232,241].

### 7.7. Supercrtical Fluid Extraction (SFE)

Here there is loading of the sample in elevated pressure vessel and extraction is done with supercritical fluid, most routinely carbon dioxide [242], with methanol included to the modifier when polar compounds are extracted [243,244] at the pressure 150–450 bar and temperature of 40–150 °C. The collection of analytes is done in a little volume of solvent or onto solid phase trap which is rinsed with solvent in a successive step. Extraction time for SFE is 10–60 min extracting 1–5 g of sample with 2–5 mL solvent usage for solid trap and 5–20 mL liquid. Advantages of SFE include fast extraction, minimum solvent volumes, elevated temperature, adequately selected towards martrix interferences, no clean up or filtration required, and extracts are concentrated in an automated system. Its major drawbacks are the numerous parameters to optimise [242].

### 7.8. Soxhlet Extraction (SE)

With the use of Soxhlet extraction, the solid sample is put in a glass fibre thimble and by employing a Soxhlet extractor, the vapour of solvents frequently seeps the sample. The extraction time is 3–48 h, extracting 1–30 g of sample with 100–500 ML of solvent. SE has an advantage of no filtration needed but its limitations are long extraction time, large solvent consumption and needed clean-up step. It also wastes cooling water and electric energy [242], and labour intensive. Notwithstanding, it has been utilized for extraction of organic compound from solid matrices owing to its elevated extraction effectiveness [245,246,247,248]. It has been utilized in compound like triclosan [249]. The extraction is investigated in three stages: rinsing, boiling, and solvent recovery [250]. There is not much work done on extracting PPCPs using Soxhlet extractor.

## 8. Clean Up and Pre-Concentration

After extraction, clean-up is occasionally expedient to minimize the limit of detection (LOD) and to reduce the interferences. Many techniques which are in use for the clean-up of PPCPs include gel permeation chromatography (GPC), columns for purifying extracts from PLE [215], and SPE for both clean-up and pre-concentration with the use of C_18_ and Oasis HLB as sorbents [190,197]. The sorbents like ENV+ Oasis HLB, Strata-X, Lichro lut C_18_ and Lichor EN have been utilised for clean-up and pre-concentration of pharmaceuticals in water [134].

## 9. Chromatographic Analysis and Detection

### 9.1. Gas Chromatography-Mass Spectrometry and Gas Chromatography-Mass Spectrometry/Mass Spectrometry (GC-MS and GC-MS^2^)

GC-MS and GC-MS^2^ remain the predominantly utilised techniques because of their higher level of sensitivity, broad accessibility in environmental laboratories, and the well-accepted electron-impact (EI) MS libraries. With suitable derivatization, GC-MS or GC-MS^2^ is a sensitive, cost-effective method suitable for regular analysis. It is also worth noting that GC-MS or GC-MS^2^ does not suffer much from the matrix effect that is often observed in electrospray ionization (ESI)-based LC-MS or LC-MS^2^ analysis [251]. However, because of the poor volatility of certain compounds, derivatization steps which focused on producing additional volatile products are needed to enhance the sensitivity of the following GC analysis. Consequently, the merits of superior sensitivity are from time to time largely nullified by losing of sample in the extra manipulation [251,252]. Excluding some neutral drugs and fragrance constituents (musks), most PPCPs are polar, non-volatile, and thermally unstable compounds that are inappropriate for GC separation. Derivatization of hydroxyl- and carboxyl-groups preliminary to GC-MS or GC-MS^2^ analysis of PPCPs has hence become a needful step. Derivatization is normally carried out by using organic reactions (e.g., silylation, methylation, and acetylation) following analytes extractions and clean up from the sample matrix. Although, the results may be affected by individual experimental parameters (e.g., reaction time, temperature, and different reactivity agents utilised). Derivatization agents are normally preferred following their reaction with the analytes or the steadiness of their yield so that hydrolysis will not likely happen [253].

### 9.2. Liquid Chromatography-Mass Spectrometry and Liquid Chromatography-Mass Spectrometry/Mass Spectrometry (LC-MS and LC-MS^2^)

LC–MS has earned vogue in the recent years, because of the sensitivity, sturdiness, and simplicity of utilisation allowed by the current API interfaces, such as atmospheric pressure chemical ionization (APCI) and electrospray (ESI) [254]. Incorporated with a novel inception of MS equipment (single quadrupole, ion-trap, triple quadrupole), LC–MS and LC–MS–MS have not just been prevalent, they are also regular techniques for respective classes of EDCs, such as natural and synthetic steroids, alkylphenolic compounds, and bisphenolic compounds.

LC–MS is an appropriate method given that no derivatization process is needed. It allows carrying out detection and quantification in one distinct step. Although, the matrix effect originated by the exorbitant concentration of matrix ionisable constituents—e.g., salts, ion-paring agents, natural organic matter, non-target pollutants—that can meddle with the ionization processes, mainly when ESI interface is employed is one of the focal disadvantages of the investigation of trace organics in composite samples by LC–MS/MS [255,256]. This reaction can be more substantial in intricate environmental samples, like sewage sludge, and it might lead to a clue prevention or improvement, resulting to little sensitivity and unreliable conclusions [255,256]. Accordingly, to make certain the accuracy of the performed technique, the assessment of the matrix effect is generally regarded in the reported literature as a bit of the method validation [232]. Furthermore, LC-MS^2^ has been more routinely employed in pharmaceutical investigation for the reason of its elevated sensitivity and its potential to establish compounds in comparison with usual LC with ultraviolet (UV) or fluorimetric detection. LC-MS^2^ permits isolation and identification of compounds which have dissimilar product ions though equal molecular mass even if they co-elute. MS^2^ detection is consequently for enlarged analytical sensitivity and selectivity in composite samples like wastewaters [257].

### 9.3. Capillary Electrophoresis (CE)

Opportunely, capillary electrophoresis has proven to be cheap, effective, and speedy separation method that has been broadly related in investigation of pharmaceuticals [258]. Analytes that are difficult to isolate by LC are frequently rectified using CE for the reason of its being selective by additives, pH tuning, and buffer concentration [259]. Sensitivity restrictions generated by short light path lengths and sample injection volumes in nanoliter may possibly be controlled by on-line pre-treatment techniques like stacking and sweeping [260,261], or off-line sample pre-treatment proficiency such as solid-phase micro-extraction (SPME), SPE, and LLE. Consequently, some CE methods have been shown to determine PPCPs in natural water [262], drinking water [263], river water [264], tap water [265], and stagnant water [266]. The major frequently used methods are GC and high-performance liquid chromatography (HPLC) with mass spectrometry (MS) detection [23,142,267]. Though, when chromatographic techniques are engaged, intricate sample pre-treatment is required and defective isolation for the comparatively polar compounds is a hitch. Furthermore, the excessive investigation cost renders these techniques less popular in standard research. CE has been told to be an inexpensive, effective, and swift separation method that has been extensively related in pharmaceutical investigations [258,268].

### 9.4. High Performance Liquid Chromatography-Mass Spectrometry (HPLC-MS)

A technique based on separation and detection was established by Andreozzi et al. [269]. The utilisation of MS as the detector vehemently restricts the option of a mobile phase. The mobile phase should be volatile, ideally not simply ionisable, and repress the interaction of analytes and silanol groups, which leads to peak tailing. Alternatively, the mobile phase should react with analytes to yield charged ions. Moreover, a separation of such chemically and structurally dissimilar compounds as b-blockers and fluoroquinolones needs a stationary phase which is both inactive (highly deactivated) and shows adequate retention time.

## 10. Detection of PPCPs

Analysis for trace EDCs has been influenced by two instrumentation techniques for the past few decades—GC with several types of detectors, including flame ionization detector (FID), electron capture detector (ECD), and MS [270,271,272]; or the LC with a broad diversity of detectors, such as MS, diode array (DAD), and fluorescence detectors [273,274]. Every instrumentation technique has its own superiority and their way of measuring EDCs is dependent on the chemical and physical properties of the analytes being investigated. Though, employing these instrumental methods has definite drawbacks given as inability to attain needed sensitivity and having deficient selectivity in several occasions. Accordingly, a significant switch was detected when more selective, seasoned and sensitive detector systems such as ion trap MS, triple quadrupole MS, orbitrap MS and time-of-flight MS was chosen by researchers. GC-MS and LC-MS-MS remains the desired instrumental methods for the investigation of EDCs, especially for pharmaceuticals, alkylphenol compounds, and estrogenic hormones in environmental samples [275].

Identification is an age-old complex and strenuous procedure due to the scope of materials possibly seen in clean water [149]. Mass spectrometric (MS) detection is currently desired because of its authentication influence and measurement to the small concentration of these pollutants in sewage sludge. Albeit single-quadrupole instruments were utilized successfully whilst LC-MS methods were initially developed to investigate residues of PPCPs [211], alternative composite mass analyzer (e.g., triple-quadrupole analyzers (QqQs)) are mostly used so the target analytes can be detected unambiguously [189,190]. Tandem mass spectrometry (MS/MS), such as triple-quadrupole (QqQ) and quadrupole time-of-flight (QqQ) are time and again used for investigation of pharmaceuticals in environmental matrices with the integration of PLE, LC, and MS or tandem MS (MS^2^). Limits of detection (LODs) at small µg/kg degree of dry weight (d.w.) can be acquired [276,277] in agreement with Diaz-Cruz et al. [186] who asserted that LODs obtained with LC-MS^2^ techniques, working in SRM mode, have been lower than those reached with GC-MS, but stated that they will also require more versatile and less complicated sample preparation.

## 11. Conclusions

Continuous use of PPCPs by both humans and animals on daily basis has hugely contributed to their persistence in the aquatic environment. Although, there is no guideline for PPCPs, the rate at which they are taken up by humans and animals in the environment is risky. They have been listed as endocrine disrupting chemicals. Wastewater treatment plants in Africa are not designed in such a way to remove PPCPs. SPE with the use of Oasis HLB and Oasis MCX have been verified to be the most efficient method of extracting PPCPs from water samples and also a means for aqueous samples clean-up. Methanol and water are considered as choice solvent for their extraction from water using liquid chromatographic technique, while ultrasonic extraction has been routinely used for sediment samples, given its low consumption of solvent unlike Soxhlet extraction. LC-MS with triple quadrupole remains the most widely used analytical method of detection because it does not require cumbersome activity like derivatization for GC-MS and has a great advantage as regards matrix effect. Although, efficient result can also be achieved with GC-MS with suitable derivatization of the compounds in spite of poor volatility of certain compounds.

## Figures and Tables

**Table 1 ijerph-16-03026-t001:** Extraction techniques of various pharmaceuticals and personal care products.

Analyte	Sample Matrix	Extraction Mode	Analysis	Concentration	Reference
Caffeine	Influent	b	m	25,578 ng/L	[24]
	Effluent	b	m	115.1	[24]
	River	b	m	34–238	[24]
Carbamazepine	Influent	c	P	2.1 µg/L	[278]
	Effluent	c	P	0.39 µg/L	[270]
	Activated sludge	c	P	0.76 µg/L	[270]
	Surface water	c	p	28.3 ng/L mean conc	[267]
Sulfamethaxazole	Influent	b	m	356 ng/L	[24]
	Effluent	b	m	22 ng/L	[24]
	Intermediate	b	m	57 ng/L	[24]
	River1	b	m	25−58 ng/L	[24]
Triclosan	Influent	c	p	1.8 µg/L	[272]
	Effluent	c	p	0.05 µg/L	[270]
	Activated sludge	c	p	0.09 µg/L	[272]
Naproxen	Biosoild	e	o	470 ng/dw	[273]
	Influent	c	p	14 µg/L	[272]
	Effluent	c	p	0.08 µg/L	[272]
	Activated sludge	c	p	1.8 µg/L	[272]
	Influent	b	m	863 ng/L	[24]
	Effluent	b	m	170 ng/L	[24]
	Intermediate	b	m	224 ng/L	[24]
	River1	b	m	18 ng/L	[24]
	River2	b	m	96 ng/L	[24]
	River3	b	m	60 ng/L	[24]
Ibuprofen	Influent	c	p	21 µg/L	[272]
	Effluent	c	p	0.06 µg/L	[272]
	Activated sludge	c	p	2.2 µg/L	[272]
	Wastewater	e	s	4.8 mg/L instrumental	[276]
Diclofenac	Sediment	j	q	<LOD-309 ± 15.3	[32]
	Influent	c	p	3.2 µg/L	[272]
	Effluent	c	p	<LOD	[272]
	Activated sludge	c	p	0.15 µg/L	[272]
	Influent	b	m	1993 ng/L	[24]
	Effluent	b	m	632 ng/L	[24]
	Intermediate	b	m	1665 ng/L	[24]
	River1	b	m	186–839 ng/L	[24]
Ketoprofen	Influent	c	p	1.5 µg/L	[272]
	Effluent	c	p	<LOD	[272]
	Activated sludge	c	p	0.02 µg/L	[272]
DEET	Influent	b	m	124 ng/L	[24]
	Intermediate	b	m	121 ng/L	[24]
	Effluent	b	m	79 ng/L	[24]
	River1	b	m	22–94 ng/L	[24]
Acetaminophen	Intermediate	b	m	233 ng/L	[24]
	Effluent	b	m	115 ng/L	[24]
	River1	b	m	93−278 ng/L	[24]
Ampicillin	Sediment	j	q	50.8 ± 2.66–369 ± 9.0	[32]
Aspirin	Sediment	j	q	212 ± 1.6–427 ± 4.47	[32]
Nalixidic acid	Sediment	j	q	117 ± 23.1–455 ± 12.2	[32]
Clofibiric acid	Influent	c	p	<LOD	[272]
	Effluent	c	p	<LOD	[272]
	Activated sludge	c	p	<LOD	[272]
Estradiol	Influent	b	m	1165 ng/L	[24]
	Intermediate	b	m	862 ng/L	[24]
	Effluent	b	m	19.8 ng/L	[24]
	River1	b	m	211−228 ng/L	[24]
Parecetamol	Influent		p	77 µg/L	[272]
	Effluent	c	p	0.18 µg/L	[272]
	Activated sludge	c	p	0.33 µg/L	[272]

a—QuEchERS; b—SPE Oasis HLB; c—SPE; d—online SPE reversed phase; e—PLE; f—MAE-Hf-L/SME; g—PHWE; h—SPE-CE; i—USAEME; j—ultrasonication; l—UPLC MS-MS; m-LC-MSMS; n—LC-DAD-MS; o—LC-UV-ESI-MS; p—GC-MS; q—LC-ESI-MS-MS; r—L.

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
