# Peer review of "Distribution and Chemical Analysis of Pharmaceuticals and Personal Care Products (PPCPs) in the Environmental Systems: A Review"

_ijerph, 2019, doi:10.3390/ijerph16173026_

Round 1

Reviewer 1 Report

This manuscript give a comprehensive description on the distribution and chemical analysis of pharmaceuticals and personal care products (PPCPS) in the aquatic system. The work make sense and provide our new sight on understanding the PCPPS. However, before accepted, there are several major comments need to consider.
1. On the introduction part: The author need to give more introduction about the pharmaceuticals and personal care products (PPCPS), for example, what kinds of these chemicals include, which is the most frequently used? What about the production of this chemicals all of the world? These information is important to fully understand them.

2. On line 62-63: Dose synthetic estrogen compounds belong to PPCPS, as our known, we consider these as typical EDCS. If not, why author mentioned it here?

3. On the Materials and Methods part: About the toxicological effects, in this work the author focus on the distribution of PPCPS on the aquatic system, so suggest he author add more toxicological studies on the aquatic organism.

4. On the distribution and mobility: Suggest the authors add the data about the typical PPCPS distribution on the aquatic system, for example, you can make a table about the PPCPS’ concentration on the aquatic system (such as surface water, sediment, aquatic organism). Actually, after I reading your paper, I did not get the PPCPS’ effective distribution information.

5. In generally, there are many kings of PPCPS, the authors need give us some effective information about this, you can choose some typical one to give detailed description. In addition, the health risk of PPCPS to the aquatic system should also added.

Author Response

The Responses to the comments are attached and uploaded

Reviewer 2 Report

This review provides a useful meta-analysis of the methods for analysis of environmental contaminants, particularly PPCPs.  The authors have reviewed a number of appropriate sources and are thorough in their discussion of the analytical methods.  The title does not reflect the content of the review (particularly use of the term "distribution"). The introduction is redundant and cursory in its scope.  The review needs extensive revision in the area of English language and style.  The introduction should be reorganized and revised to reduce redundancy and to improve clarity.  Headings should be reviewed and revised to better reflect section content.  Some sections would benefit from brief introductory remarks.  In-text citations need to be reviewed and organized; in some cases, the reference citations are incomplete. 

Author Response

The  responses to the reviewer's comment is attached and uploaded

Round 2

Reviewer 2 Report

The authors have addressed some of the concerns of the original reviews of this manuscript.  I appreciate their response.  I continue to have concerns, however, about the some of the content and, particularly, the manuscript needs revision of English language use and style.  I suggest the authors use a professional English editing service.

In addition to general editing throughout, I have the following concerns about the manuscript:

·         Work on spelling and word choice, a few examples (not comprehensive) include:

Line 42: caffeine spelling

Line 43: “antimoral”—what is this referring to?

Line 268: “antibodies” (used twice)—this should be “antibiotics”

Lines 433 and 434: italicize scientific names

Line 448: “disclosure” should be “exposure”

·         Work on content/organization (again, these are examples and not a comprehensive list):

Some sections are redundant, particularly in the introduction

Line 47: are lipid regulators pharmaceuticals rather than PCPs?

Lines 72-76: are not relevant to the rest of the paragraph

Lines 103-107: more relevant in the paragraph starting with lines 77-78

Lines 120-123: discussion of DDT and POPs is not relevant to the PPCP discussion

Lines 142-144: provide a good conclusion to the introduction; however, the introduction then continues with the “Examples of different kinds of PCPs.”  Move this section (lines 146-196) earlier in the introduction.

Lines 203-207: are not relevant to the paragraph

Lines 332-334: are not relevant to the paragraph

Lines 370-372: are unclear

Line 470: remove “details of various” as Table 1 lists extraction techniques but not the details of each; cite table 1 when discussing PPCP concentrations.

Line 753: earlier in the manuscript PPCPs are said to not be bio-accumulative so the statement in this line is confusing.

Lines 761-762: this information is not discussed in the body of the manuscript.

·         Work on references.  Some examples include:

Line 110: number the EPA reference

Line 133: number the Richardson et al. reference

Line 141: a reference is needed for this statement

Line 258: number the Lange et al. reference

Line 421: a reference is needed for this statement

Line 781:  First author’s name should be Ebele, AJ

Line 1495: number reference and ensure it is cited in the manuscript

·         Work on headings/organization.  A few examples:

Line 335 and 336: since there is only one subheading, without any introductory text, include the subheading as the heading.

Line 389: subheading number is listed twice

Line 635: heading does not reflect the content of the section; use the subheading as a heading instead.

Lines 643-644: Make this one heading; then sub- subheadings will not be necessary.

Line 720: Should this be read “Detection of EDCs”?

Author Response

Manuscript ID: ijerph-534226

Responses to reviewers’ comments

Reviewer #2

Introduction

Comment: Line 42: caffeine spelling

Line 43: “antimoral”—what is this referring to?

Line 268: “antibodies” (used twice)—this should be “antibiotics”

Lines 433 and 434: italicize scientific names

Line 448: “disclosure” should be “exposure”

Response:   Spellings of “caffeine” corrected as evident in line 43.

“Antimoral” has been deleted in line 43.

Line 257-258 has been reworded and the word “antibodies” has been changed to “antibiotics”.

Scientific names have been italicized in line 419 and 420.

“Disclosure” has been replaced with “exposure” in line 434.

Comment: Work on content/organization (again, these are examples and not a comprehensive list):

Some sections are redundant, particularly in the introduction

Line 47: are lipid regulators pharmaceuticals rather than PCPs?

Lines 72-76: are not relevant to the rest of the paragraph

Lines 103-107: more relevant in the paragraph starting with lines 77-78

Lines 120-123: discussion of DDT and POPs is not relevant to the PPCP discussion

Lines 142-144: provide a good conclusion to the introduction; however, the introduction then continues with the “Examples of different kinds of PCPs.”  Move this section (lines 146-196) earlier in the introduction.

Lines 203-207: are not relevant to the paragraph

Lines 332-334: are not relevant to the paragraph

Lines 370-372: are unclear

Line 470: remove “details of various” as Table 1 lists extraction techniques but not the details of each; cite table 1 when discussing PPCP concentrations.

Line 753: earlier in the manuscript PPCPs are said to not be bio-accumulative so the statement in this line is confusing.

Lines 761-762: this information is not discussed in the body of the manuscript.

Response: Lipid regulators have been listed as pharmaceuticals and not PCPs in lines 45.

Lines 72-76 has been deleted.

The statements in lines 103-107 have been taken to lines 83-86.

Lines 120-123 have been deleted.

Lines 146-196 have been moved up to lines 136-186, and conclusion has been made in lines 167-189 as recommended.

Line 203-207 have been deleted.

Lines 332-334 have been moved down to lines 334-336, corroborating the fact that PPCPs can also be detected at low levels at non-point sources.

Lines 370-372 have been deleted as recommended.

The words “details of various” in line 470 have been deleted in 456.

The “bioaccumulative” in line 753 has been deleted in line 729.

Lines 761-762 have been modified in 737-738.

Work on references.  Some examples include

Comment:

Line 110: number the EPA reference

Line 133: number the Richardson et al. reference

Line 141: a reference is needed for this statement

Line 258: number the Lange et al. reference

Line 421: a reference is needed for this statement

Line 781:  First author’s name should be Ebele, AJ

Line 1495: number reference and ensure it is cited in the manuscript

Response: Reference in line 110, now 108 has been numbered.

Line 125: Richardson et al has been numbered.

Line 134: The reference of line 141 has been provided.

The reference in line 258 has now been provided in line 247.

Line reference in 421 has been provided in line 407.

Line 781 now 757: The first author’s name has been corrected.

Line 1495 has been deleted.

Work on headings/organization.  A few examples:

Comment: Line 335 and 336: since there is only one subheading, without any introductory text, include the subheading as the heading.

Line 389: subheading number is listed twice

Line 635: heading does not reflect the content of the section; use the subheading as a heading instead.

Lines 643-644: Make this one heading; then sub- subheadings will not be necessary.

Line 720: Should this be read “Detection of EDCs”?

Response: Line 335-336 have been merged as one heading in line 324.

The subheading in line 389 has been corrected (now in line 360).

Line 635 now 621 has been deleted and the heading modified.

Line 643-644 has been merged with the heading in line 628.

Line 720 has been corrected to read “Detection of PPCPs” in line 696

English language editing

Comment: The authors have addressed some of the concerns of the original reviews of this manuscript.  I appreciate their response.  I continue to have concerns, however, about the some of the content and, particularly, the manuscript needs revision of English language use and style.  I suggest the authors use a professional English editing service

Response:

This work has been edited by proficient English language scholars

Round 3

Reviewer 2 Report

The authors have addressed specific concerns raised in previous reviews.  The identified concerns were representative of broader issues with the manuscript.  There continues to be areas of the manuscript that would improve with organization and English language revision.